# Preliminary Considerations for Crime Scene Analysis in Cases of Animals Affected by Homemade Ammonium Nitrate and Aluminum Powder Anti-Personnel Landmines in Colombia: Characteristics and Effects

**DOI:** 10.3390/ani12151938

**Published:** 2022-07-29

**Authors:** Carlos Jaramillo Gutiérrez, Gustavo Farías Roldán, Krešimir Severin, Ubicelio Martin Orozco, Pilar Marín García, Víctor Toledo González

**Affiliations:** 1Criminal Investigation Program, Law School, University of Medellín, Carrera 87 #30-65, Medellín 050026, Colombia; cajaramillo@udem.edu.co; 2Department of Animal Pathology, Faculty of Veterinary Sciences Physiopathology and Toxicology Laboratory, University of Chile, Avda. Santa Rosa 11735, 88.208-08 Cas. 2 Co. 15, La Pintana 6640022, Chile; gfarias@uchile.cl; 3Department of Forensic and State Veterinary Medicine, ForesicLAB, Faculty of Veterinary Medicine University of Zagreb, Heinzelova 55, 10 000 Zagreb, Croatia; severin@vef.unizg.hr; 4Laboratory of Anatomic Veterinary Pathology, Department of Veterinary Science, Institute of Biomedical Science, Autonomous University of Ciudad Juárez, Av. Benjamin Franklin No. 4650, Zona PRONAF, Ciudad de Juárez 32315, Mexico; umartin@uacj.mx; 5Department of Anatomy and Embriology, Faculty of Veterinary, Complutense University of Madrid, Avenida Puerta de Hierro, s/n., 28040 Madrid, Spain; pilmarin@vet.ucm.es; 6CINQUIFOR Research Group, Department of Analytical Chemistry, Physical, Chemistry and Chemical Engineering, University of Alcalá, Ctra. Madrid–Barcelona km 33.600, 28871 Alcalá de Henares, Spain; 7University Institute of Research in Police Sciences (IUICP), University of Alcalá, Colegio Máximo de Jesuitas, Calle Libreros 27, 28801 Alcalá de Henares, Spain

**Keywords:** homemade antipersonnel landmine, landmines, biodiversity, crime scene analysis, veterinary forensic medicine

## Abstract

**Simple Summary:**

Anti-personnel landmines are a major problem in countries that are subject to internal conflicts of a military or public order nature. They also continue to be a great threat to the population and biodiversity, even in post-conflict stages. Those most often used by armed groups are simple or homemade antipersonnel landmines that are designed without any type of technical regulations and standardized production systems. Their low-cost manufacturing and the use of easily accessible explosive substances for agricultural use, such as ammonium, have allowed their indiscriminate use, turning them into a huge public health problem. They are only detected when people or animals activate them, because they do not contain any materials that are detectable by traditional means. The scant literature on these artifacts focuses on injuries caused to humans, and only incidentally on field work. The objective of our study was to describe the behavior of a controlled explosion of a homemade antipersonnel landmine, and to verify the effects caused by the explosion on fauna and the environment. The results enable us to provide guidelines that may be implemented during field investigations, in which forensic veterinarians and related disciplines participate.

**Abstract:**

During the armed conflict in Colombia, homemade improvised antipersonnel landmines were used to neutralize the adversary. Many active artifacts remain buried, causing damage to biodiversity by exploding. The extensive literature describes the effects and injuries caused to humans by conventional landmines. However, there is considerably less information on the behavior and effects of homemade antipersonnel landmines on fauna and good field investigation practices. Our objectives were to describe the characteristics of a controlled explosion of a homemade antipersonnel landmine (using ammonium nitrate as an explosive substance), to compare the effectiveness of some evidence search patterns used in forensic investigation, and to determine the effects on a piece of an animal carcass. The explosion generated a shock wave and an exothermic reaction, generating physical effects on the ground and surrounding structures near the point of explosion. The amputation of the foot in direct contact with the device during the explosion and multiple fractures were the main effects on the animal carcass. Finally, it was determined that finding evidence was more effective in a smaller search area. Many factors can influence the results, which must be weighed when interpreting the results, as discussed in this manuscript.

## 1. Introduction

Colombia is one of the world’s most biodiverse countries affected not only by illegal animal trafficking, but also by armed conflicts [1]. Like humans and domestic animals, various species of wild animals have been and continue to be victims of the indiscriminate use of improvised non-conventional anti-personnel landmines [2]. This is also a reality that occurs in other countries [3]. Many of them were planted more than 50 years ago, and in the 1990s [4], during internal armed conflicts between illegal armed groups and the state [5], and remaining hidden and active beneath the surface [3]. These mines do not distinguish between targets, and injury from anti-personnel mines continues even after the conflict has ended [3]. An anti-personnel landmine (AP) is an explosive device buried in the soil [5], very close to the surface, which causes indiscriminate damage when a person or animal steps on them [6,7]. Landmines are designed to maim rather than kill, generating serious physical and psychological damage [8]. In general, an improvised explosive device has two very basic components: the main-charge explosive and a fusing system [9]. However, there are numerous types of anti-personnel, varying in size, construction, and explosive type [8]. The classification of anti-personnel landmines depends on the type of activation, and whether victim- triggered or remotely operated, amongst other aspects [10]. Remote activation devices have metal parts, making their detection easy, but the first type may or may not have low metal content [11]. This means that there are detectable and undetectable non-conventional AP landmines [4]. Their construction requires low cost, minimal technical knowledge, basic tools for assembly, and they can be triggered by pressure, tension, and release [11]. Armed groups have created various strategies to construct improvised devices using non-traditional components, such as non-metallic containers and syringes as fuses [10], making their identification difficult using conventional methods and commercial devices [10]. These are called undetectable homemade AP landmines [4,10]. Since their discovery occurs coincidentally when they are triggered by direct contact with the victim [4], they constitute a major problem for public health [12] and the ecosystem [10,13,14]. Antipersonnel landmines are considered the worst kind of global pollution [15]. In contrast to conventional explosive devices, homemade AP landmines do not have a manufacturing data sheet which makes it impossible to predict the real effect of their explosion. This is because the proportion of each element used in the final mixture of the explosive substance or explosive charge will depend on availability, the manufacturer, and the degradation of its components [4]. Even so, they can remain active for a long time, not discriminating between their potential victims, and patiently waiting to be triggered. Landmines are generally classified as blast [5] or fragmentation mines [5,16]. Blast mines are buried at a shallow depth and their mechanism is generally triggered by the pressure exerted by the victim stepping on the mine [5]. Both these explosive devices cause the affected object (usually the victim’s lower limb) to explode into fragments in an upward direction [17,18]. On the other hand, fragmentation mines disperse fragments stored inside the container (splinters) (e.g., metals, stones, nails, glass, etc.), radially outward at high speed [17]. In general, the detonation of an explosive initiates a wave that propagates through the explosive, causing an instantaneous chemical reaction, converting the explosive into a hot gas at high pressure (the products of detonation) [19]. This expands, forcing the volume it occupies to form a layer of compressed air (blast wave) in front of it, which contains most of the energy released by the explosion. This means that the blast wave is formed by a sudden release of energy [19]. The over-expansion of the detonation products results in the development of a sub-atmospheric pressure phase, creating a partial vacuum and air suction. Strong suction winds accompany this process, transporting debris long distances from the source of the explosion [20]. Search methodologies in field work in cases of explosive devices use conventional explosive devices and humans in order to determine the type of seat of the explosion of the explosive device, the explosion trace, the explosive substance, the charge mass, and the damage caused [9,21]. However, information about the effects and damage caused by homemade AP landmine explosions and procedures at the crime scene, when animals are involved, is almost nonexistent. This study aims to discuss these procedures and the effects that homemade AP landmines could generate on the part of an animal carcass and its immediate environment. Finally, it provides some guidelines for the fieldwork of first responders, researchers, veterinarians, etc., in order to define the perimeter and choose the best evidence search pattern, when dealing with cases of this nature. 

## 2. Materials and Methods

An improvised, homemade AP landmine designed to be triggered by pressure was used for this study. It was manufactured in the laboratory for forensic analyses of terrorist attacks by one of the authors of this study, a specialist in explosive devices from the Universidad de Medellín, Colombia. Likewise, the experiment was carried out in a place officially designated for such studies (PICMA; Program in Prevention and Research against Antipersonnel Mines) at the University of Medellin, and under strict safety standards established by the center and experts. The specific zone of the experiment (closed by yellow tape) is located in a valley surrounded by hills (See below). For the study, we had the collaboration of students from the PICMA group and the Criminal Investigation program of the Faculty of Law of the University of Medellín. During first step of experimentation, 20 collaborators were positioned at different points around the experimental area, more than 40 m from the epicenter of the explosion. The soil of the experimental field is classified as black clay. In general, black-gray soil is nutrient rich, and but it retains moisture due to waterlogging caused by drainage problems. The ground surface was firm and slightly wet. In addition, vast vegetation covers the experimental field.

### 2.1. Homemade AP Landmine Design

Figure 1 shows the materials and their arrangement inside the container. A 100-cc hypodermic propylene syringe (Precision care^®^, Bogotá, Colombia) was used as a fuse in the manufacture of the device. Aluminum foil and a copper wire covered the tip of the plunger. This plunger functioned as a switch since its electrode made contact with another electrode located at the lower end of the barrel when the pressure dropped, closing the circuit of energy generated by the power (GP 1604GLF 6F22 9V 300mAh Green Cell^®^ Battery, Rainford, UK) necessary for activation of the electric detonator. The power source is also known as the exploder. This energy generates a piezoelectric effect on a crystal contained inside the detonator, in our case a ferronickel crystal. Due to its properties, this crystal is used to produce the electrical energy for the initiation of the detonators. Consequently, its activation will initiate the explosion of the explosive substance. Tin solder was used to connect the 18-gauge copper wires (diameter 1.02 mm) to the power source and detonator. Due to the impossibility of activating the detonator by direct pressure on the fuse, the mechanism was slightly adapted. For this purpose, an electric charge was sent directly to the detonator using copper cables connected to an external energy source located more than 40 m away from the explosive device. The explosive substance used was a mixture known as R1, composed of ammonium nitrate, black aluminum, and sawdust, according to previous publications [4]. The container was a 3-inch diameter, 1.8 mm thick, 12 cm long PVC pipe (Gerfor Ind. Col. PVC Sanitaria^®^, Medellín, Colombia). The amount of explosive substance was one tenth of the total capacity of the container (100 g). The container was waterproofed against moisture by covering it with Tesa^®^ silver paper adhesive tape (Barcelona, Spain). Finally, a lid of the same material and thickness was perforated to fix the syringe to the device. The syringe was fixed to the cap with tar, preventing it from moving as the plunger moved inside the barrel when pressed. The total weight of the artifact was 450 g. For ethical and safety reasons, only the compounds used for the explosive charge will be indicated, but not the percentages of them used in the mixture, nor the processing method. The artifact was buried leaving only the fuse visible on the surface, as shown in Figure 2D.

### 2.2. Biological Sample

To simulate the effects of homemade AP landmines on a live animal, the fresh rear end of the carcass of a dead calf was used that had died from dystocia during parturition. The dead calf was removed by fetotomy 24 h before the experiment by a local veterinarian. The part of the animal donated by the owner for scientific purposes included the hind limbs, the pelvis, and the viscera contained between the abdominal and pelvic cavities (e.g., the small intestine and colon, bladder, spleen, and kidneys) (Figure 2A). Due to the natural cause of death of the animal, unrelated to the present study, ethical review and approval were not required. To simulate a standing animal, the carcass was positioned using a wooden stick as a means of support (Figure 2). The cranial portion of the part of carcass was turned to the north of the experimental field, the caudal side to the south, the right side to the east, and the left side to the west. The explosive was buried a few centimeters from the surface under the right foot of the animal, simulating what would happen when it stepped on the fuse, triggering the detonation (Figure 2E). A thin layer of about 2–3 cm of soil was interposed between the foot and the artifact. To demonstrate the effect and scope of the explosion, some anatomical reference points and distances from explosive device were registered (Figure 2B–D).

To the naked eye, there was no evidence of the presence of other animal species around the of part of the carcass before the explosion.

### 2.3. Considered Variables

The variables for the study were the presence/absence of sound, odor, heat, smog, vibrations, a crater with a defined epicenter or none, the animal or the part of an animal, the landmine fragments, and the direct effects on the animal. In addition, the crater, the area around crater, the area beyond the immediate area, witness materials, and the effects caused on the part of the animal were described. Temporal variables, such as sound, odor, heat, smog, and vibrations (temporal variables), were recorded by direct empirical observation to determine their presence/absence (see below). The dimensions of the crater (length and width) were considered by taking the average of three measurements made at different points. The description of the effects on the part of the animal carcass was carried out by simple dissection and observation.

### 2.4. Radius of the Action of the Explosion on the Dispersion, Distance and Distribution of Projected Fragments

To determine the dispersion patterns and extent to which the evidence (biological or not) was projected by the explosion, a circular perimeter was established using a thin white paper fence 1 m wide (Bond^®^ 75 Grs, Bogotá, Colombia) within a radius of 4 m around the homemade AP landmine (Figure 3). It was held up by wooden pillars. The height from the ground to the lower and upper edge of the fence was 0.4 m and 1.4 m, respectively. The space under the paper fence was covered by vegetation. In order to show the effect of the explosion on the surface of the ground (under the fence), a wooden box weighing 1.8 kg was positioned 2 m to the left of the part of the animal carcass.

To determine the trajectory of the expelled part of carcass and other possible fragments during the explosion, three video cameras (Legria HF G40^®^ and Canon EOS 1200D^®^, Canon Europa N.V, Bovenkerkerweg, The Netherlands), positioned to the north, south, and east of the epicenter were used. The northern camera recorded the very first explosion in the central scene with a frame rate of 50.00 frames/second. The western and northern cameras registered a panoramic view of the explosion with a frame rate of 29.97 frames/second. The cameras were mounted on a Hama tripod (Hama Technics, S.L., Barcelona, Spain), 40 m from the point of detonation. Video clips were shot in MP4 format. Some snapshots are shown in the results section. 

### 2.5. Evidence Research

The effectiveness of four search patterns commonly used in crime scene investigations (line and strip, grid, zone, and spiral) and previously described by other authors [9,22] was evaluated. The crater was considered the central point of the scene. Each strip had a width of 2 m and each zone covered an area of 4 m^2^. The same entry and exit point were defined for each line and strip pattern. This means that the search was performed in both directions. The time established for each search pattern was one hour. The evidence was collected at the end of the experiment just for registration purposes. Different groups of criminalists conducted the search separately and independently, with natural light between 14:30 and 19:45 h. They entered in the field in a coordinated manner to avoid contact with the previous groups after the explosion. The line and strip (multiple personnel) and zonal research were carried out by a group of four professionals. Two different professionals carried out the eccentric and concentric turn patterns, and a single searcher carried out the other patterns. After a pattern of lines and individual stripes, the multiple pattern was carried out. This was followed by the circle pattern, the grid pattern, and finally the zonal pattern. The experiment was carried out on a clear and dry day, with an average temperature of 21 °C during the explosion, and light winds from west to east. A fiberglass measuring tape (Stanley^®^, New Britain, CT, USA) was used for measuring and distance recording. Measurements were made using the International System of Units (SI).

## 3. Results

### 3.1. Detonation Characteristics and Transitory Evidence

Figure 4 shows the explosion caused by our homemade AP landmine, generating a high-pressure shock wave, moving at supersonic speed. In addition, it was possible to observe a short flash (“fireball”) in the center of the explosion (exothermic reaction) (Figure 4A). At the same time, it was possible to hear an intense sound of short duration, accompanied by vibrations and smoke. The smoke changed color as it spread and dispersed from the point of the explosion (Figure 4). The smoke remained suspended in the air for 5 min. Collaborators positioned towards the eastern side of the field (i.e., downwind from the field) perceived an unnatural chemical odor in the air (e.g., ammonia) mixed with smoke. The involuntary vibration of the cameras after the explosion indicated that the shock wave propagated radially and beyond 40 m from epicenter of the detonation. Solid fragments (especially soil) were projected rapidly in all directions (radially), while the smoke traveled upward and then moved slowly in a west-east direction (Figure 4D). None of the collaborators felt any heat after the explosion. 

### 3.2. The Explosion’s Effect on Elements Present in the Blast Radius of Action

The explosion caused the deformation and projection of the soil around the detonation point, generating a crater with a clear epicenter. The bottom of the crater had a flat, mostly circular appearance. The edge of the crater at the surface was irregular and poorly defined along the entire crater perimeter (Figure 5A and Figure 5B, respectively). The average diameter of the crater at ground surface level was 1.6 m, and its depth recorded at the center of the crater was 0.51 m (Figure 5C and Figure 5D, respectively). The dimension of the bottom of the crater was 60 cm. That is to say, the shape of the crater largely resembled a cone. Vegetation located up to one meter around the perimeter of the crater was crushed in the opposite direction to the detonation point. The wooden box, although it was overturned and suffered slight damage to its structure, did not show a significant displacement from its original position (Figure 5E). Some particles of wet soil could be observed on the box. During the explosion, the part of the carcass was projected vertically but with a certain degree of inclination towards to the western sector of the field (Figure 5F). During its displacement, numerous turns could be observed until it fell to the ground. Much of the fence positioned on the northern and eastern sides was damaged or detached from its supports. In contrast, the fence placed to the west (the left side of the part of the carcass) and south suffered less damage (Figure 5G and Figure 5H, respectively). No evidence of the part of the carcass or explosive device were found in the crater.

On the entire fence, it was possible to verify the projection of particles of soil, of variable size, and distributed vertically and horizontally along the surface in a heterogeneous way. Most of these soil particles were found on areas of the fence that remained intact after the blast. Some soil sediments contained more water than others (Figure 6A,B). Others were projected above the fence and propelled over a distance of 40 m (recording points). 

### 3.3. Research Patterns

The most effective methods used in order to find evidence caused by the explosion was the zone pattern. This method allowed us to find a greater number of elements projected by explosion, especially particles of soil scattered all over the field. The same quantity of fragments belong to the part of the carcass and the homemade AP landmine were found using the other research patterns. In decreasing order of effectiveness, the patterns were as follows: the zonal pattern followed by linear and strip (multiple personnel), grid patterns, linear and strip (one person), and spiral (eccentric and concentric turns). 

### 3.4. Physical Evidence Found during the Search Process

The appearance of the evidence resulting from the explosion is shown in Figure 7. In addition to the aforementioned crater and particles of soil in the paper fence, it was also possible to find numerous particles of soil propelled from the epicenter of the explosion, scattered all over the field. This was more evident in the northern, southern, and eastern sectors of the field, that is to say, to the front, back, and right sides of the original position of the part of the carcass. Figure 7D only shows one of them. The only fragment of the container of the landmine, viscera, and the pieces of the part of the carcass were found mostly to the left side of the animal (Figure 7A–C,E). On the container fragment, it was possible to perceive a faint smell of fertilizer. 

Figure 8 shows the distribution of the evidence found on the experimental field. The distances between each item and the epicenter of the crater are summarized in Table 1.

The elements/evidence that reached the greatest distance after the explosion were pieces of clay soil. They were small, dark, and had an irregular shape. These were located more than 40 m away (not shown). The most distant fragments of the part of the carcass and explosive device were located more than 14 and 19 m away from the epicenter of the explosion, respectively (Table 1). The largest portion of the part of the carcass was located at a shorter distance than the fragments of tissues (smaller pieces). Many of the materials used in the manufacture of the explosive device could not be found, at least by the naked eye. No biological evidence was found on the paper fence, wooden box, or the fragment of the AP landmine container. Finally, none of the biological or non-biological evidence showed any incandescent elements on their structure.

### 3.5. The Effects of the Explosion on the Part of the Carcass

External observation showed the total amputation/avulsion of the structures belonging to the distal bony portion of the right limb (metatarsals and phalanges, hooves, tendons and ligaments) (Figure 9A,B). Part of the skin, tendon fibers, and nerve branches, visibly damaged, remained in place after the explosion (Figure 9C). The distal part of the right tarsal bone remained intact. In contrast, the left limb showed only slight skin damage over most of its surface. Muscle destruction/laceration and separation of the facial planes of the amputated limb and the perineal and genital region were also evident (Figure 9D). The tissues that were especially badly damaged were contaminated with impregnated remains of soil and grass fiber. It was not possible to find macroscopic remains of the AP landmine incrusted or incandescent on the tissues. An intense odor of fertilizer was prominent at the detonation point (right foot) below the skin that covered the portion of the amputated foot combined with a perceivable odor of tar. Other areas of soft tissue damage had a slight burning odor. A deeper examination of the part of the carcass showed the complete bilateral destruction of the articular capsules in the sacroiliac and coxofemoral joints. In addition, the complete destruction of the articular capsule in the tarsometatarsal joint on the right limb was evident. The articular capsules on the femorotibial and tarsocrural joints showed partial bilateral destruction. The left tarsometatarsal and left metatarsal phalangeal joints remained intact. On the skeletal system, it was also possible to recognize the bilateral separation/luxation of the coxal bones (ischium, ileum and pubis) through their growth fronts (cartilages). In fact, the coxal bones showed no evident fractures, but the coxal cartilages were absent. After dissection, the right femur displayed a transverse fracture in the proximal metaphysis. The left femur showed separation/luxation of the proximal epiphysis from the growth plate (Figure 9E). Only small parts of the right and left heads and trochanters were found to be separated into the surrounding tissue. The right tibia presented a transverse fracture on the distal metaphysis, while the left tibia showed separation/luxation of its distal epiphysis from the growth plate (Figure 9F). In the femur and tibia, the distal or proximal epiphysis, separated from the diaphysis, displayed comminuted fractures (Figure 9G,H). The metatarsal bone that remained intact (left) presented a transverse fracture in its distal metaphysis, while its distal epiphysis, separated from the bony body, also showed multiple comminuted fractures (Figure 9G,H). It was not possible to identify traces of growth plate at the dislocation points. The diaphysis of the long bones and the patellae, still present in both extremities, showed no visible fractures. Finally, most of the peritoneal viscera still contained within the abdominal cavity before the explosion were detached (Figure 9J), except for the right kidney, the bladder, and its ligaments. The right kidney was found out of its normal position. The ribs and skin covering the abdominal portion showed no visible alteration.

## 4. Discussion

In order to know what to look for, where to look, how to look, and how to protect evidence that demonstrates the use of homemade AP mines, it is essential to know their behavior after their detonation, as well as scope and effects.

### 4.1. General Aspects of the Effects of the Explosion

Our study demonstrates that a home-made AP is capable of detonating and exploding, generating a shock wave and releasing heat (exothermic reaction). That is to say, it generates detonation products. It is also capable of fragmenting and projecting elements, generating various types and degrees of damage/effects. This “homemade AP behavior after it explodes” is the same as that reported for explosions of conventional and improvised antipersonnel mines (IEDs) [8,9,21]. In particular, we will use the term explosion to refer to an effective detonation, and we will only use the concept of detonation when we refer to the products generated by the explosion. Many references speak of detonation as a synonym for explosion. However, the activation of a detonator (detonation) does not always lead to an explosion. The sequence of events that generate an explosion are describe in the methodology. The effects of the explosion on the part of the carcass, the wooden box, and the paper fence, together with the distance and dispersion pattern of the particles of soil projected onto the perimeter fence and the experimental field, allow us to understand the displacement of the blast wave. If, in addition, we consider the vibration of the cameras after the explosion, we can be sure that it spread radially, vertically and horizontally, over 40 m in distance and at least 15 m in height. This form of projection does not differ from previous reports [9]. It is possible to point out that the magnitude of the effects caused to these structures, by the blast wave and/or the projected elements, is related, among other factors, to their distance from the epicenter of the explosion. That is, less damage was generated as we moved away from the point of the explosion. It has been reported that in an open environment, the blast wave dissipates rapidly and at a predictable rate as it expands over time [8,23] and distance [21]. The highest speed of the blast wave would be reached at the point of explosion [19] and the effects caused during its displacement depend on the medium through which it travels [24]. In our study, the effects are evident at various points on the part of the carcass, although the largest ones were concentrated on the structures positioned at the point of explosion and others very close to it, as we will discuss later. Fragments projected by hot air can also affect structures near or more distant from the point of explosion, such as those found in our study. It should be noted that we will speak of the effects and not of injuries found on the part of the carcass, since we did not use a living individual that would present with a vital reaction by its tissues in response to the injury. Consequently, it is possible to infer that our results were caused by a combination of the blast wave, hot air and projected fragments, although in different ways. Some facts reinforce this statement. On the one hand, numerous particles of earth and grass fibers, projected onto the perimeter fence, were only visible on the intact areas of the fence. In contrast, the most affected areas did not show visible particles of soil on their surface. Given the small size of the soil particles found on the fence, we are inclined to believe that the destruction it underwent was caused mainly by the blast wave and not by the projection of soil fragments. There was also no evidence of fragments of the part of the carcass or the explosive device projected onto the fence. The same was true of the wooden box whose position changed slightly, and only two small particles of soil were projected onto it. Specialized reports indicate that the characteristics of the projected elements play an important role in the effects they cause [21], as we will discuss later. On the other hand, the discovery of a small fragments of container after the explosion shows us the degree of destruction/disintegration that 100 g of explosive charge (R1) can cause. The disintegration of the home-made AP would greatly reduce the possible effects caused by the projection of its fragments, especially in areas far from the point of explosion. In other words, if the effects found, especially on the part of the carcass, were caused by fragments of the artifact, these must have been generated during the first few milliseconds after the explosion and very close to it, while its components disintegrate due to the blast wave and/or or exothermic reaction. The most obvious visible effects occurred at the point of explosion, with the creation of a crater. The dimensions of the crater and the presence of particles of soil with high water content, distributed over the fence and the experimental field, show us the capacity of the blast wave to remove and project material, even from deep areas. That is, the wave also propagated downwards. Previous studies [25,26] indicate that the effect and/or behavior of the blast wave on the ground and the particles removed and projected from it depend on its structural characteristics. In our case, the projected soil particles did not cause visible damage to the wooden box or the paper fence. This is probably due to the high degree of pulverization or thinning of the particles expelled after the explosion, and their distance from the crater, as indicated above. However, and according to these last authors, soil with high water saturation (like ours) would concentrate the hot gases, generating a tunnel effect that would project the soil in a vertical direction. In addition to this, we must consider that the artifact was buried close to the surface of the ground. Since the ground offers much more resistance to movement than the air, the expansion of hot gases quickly takes the path of least resistance, and the energy and force of the explosion is directed upwards [19]. Both variables could explain, at least in part, the differential effects caused to different points of the part of the carcass and, especially, explain the amputation of the right foot positioned on the explosive device during its detonation (see below). Although the blast wave and hot air were projected and propelled elements all around, their effect on the environment was not homogeneous. The effects caused to the fence, located on the left side (western sector) and behind the animal (southern sector), were considerably less than other points. In the first case, it is likely that the part of the carcass, projected vertically after the explosion, would have had a protective effect (shield), absorbing most of the energy during the explosion, preventing further expansion of the wave to that side, as stated above. This could also explain the decreased effect of the blast wave on the wooden box and the low level of projection of soil particles onto both structures. In the second case, the opening of the fence behind the animal would have created a free escape route for the blast wave, avoiding overpressure in that area, and consequently greater destruction (Figure 5G). This hypothesis is supported by the greater number of particles of soil distributed over the sector of the experimental field to the right-hand side of the part of the carcass (eastern sector of the field) and not to the left (western sector of the field). Most of the fragments of the animal and the fragment of the explosive device were found in the latter. This shows that the obstruction at the origin of the blast wave influenced the direction, shape and strength of the wave, as indicated in other publications [21], preventing the uniform expansion of the blast wave. Another fact, mentioned superficially above, was the presence of light that denoted the exothermic reaction of the explosion. The reaction generated a high-pressure heat wave pushing the surrounding soil, compressing it, and displacing it away from the center of the explosion [19]. As we pointed out above, this wave would be concentrated at a point in clay soils, projecting elements vertically. The intense smell of burning, the large amount of grass fibers adhering to the inner surface of the skin at the point of amputation, and the absence of incandescent fragments and/or visible burns on the part of the carcass in areas far from the point of explosion reinforce this hypothesis. Normally in an AP mine explosion, thermal injury is usually limited to the region closest to the explosion, and its severity depends on the exposure time [27]. A “fireball” was evident during the explosion, although it was very short-lived. Despite the short time that the part of the carcass was exposed to heat, the high level of artifact destruction and the effects on the part of the carcass denote that the temperatures reached were high. In fact, it is indicated that short duration explosions are capable of generating very high temperatures [19] and that the heat generated can cause burns to people close to the point of explosion [21]. Unfortunately, we do not have the instrumentation to determine the temperature reached. The smoke had interesting characteristics from the investigative point of view. As time passed from the moment of the explosion, it dissipated. The particles of earth, which were also dispersed, contributed to the darker color of the smoke at the beginning of the explosion, together with the chemical reaction of the explosive substance. The smoke spread in the same direction as the wind, carrying with it an intense smell of fertilizer that lingered for several minutes. Only some of the coworkers, located in the opposite direction to the wind, perceived this effect only slightly and for a few seconds. Recognizing this “transient effect” at a crime scene is important to infer some degree of the temporality of what happened, as other authors point out [22], that is to say, how long ago the explosion occurred. In addition, it could guide us towards the source of the explosion, if we feel a current of air from the front bearing that smell. Although this procedure does not have a completely scientific basis, it is useful in investigative practice. In fact, non-human animals use this “technique” for hunting or camouflage purposes [28]. Other transient effects may include the noise and vibration caused by the explosion, as well as the smell of the fertilizer present in the fragment of the antipersonnel mine container and the part of the carcass. The latter would be useful to determine, at least qualitatively, as an unnatural chemical odor in the environment near the explosion. 

### 4.2. Effects on Part of Carcass

Before discussing the effects of the explosion, it is important to consider and/or highlight two aspects of the experimental animal model used in the study for better discussion, comparison and interpretation of the results. First, we used part of the carcass of a calf that died before birth, due to dystocia and not a complete carcass. These factors represent the intrinsic variables of the morphological and physiological character of the model used (e.g., degree of ossification, weight, nutritional age) that explain, at least some of the results that will be discussed later. Second, the abdominal cavity included in the part of the animal carcass used in the experiment was open, leaving its internal organs visually exposed and considerably unprotected to the action of external agents (e.g., direct or indirect impact, trauma, etc.). 

Injuries caused to human victims by explosions of antipersonnel mines (conventional and improvised) have been extensively reported [8,19,23] to name a few. In contrast, studies of the behavior and effects of the explosion of homemade antipersonnel mines are very limited [4]. Comparatively, despite the fact that the device used in our study was made of light materials, with no splinters inside and with a tiny amount of artisanal explosive (R1), our results show great consistency with these reports, although there are certain nuances that we will discuss later. By convention, blast injuries are classified according to the mechanism by which they occur [20,29] or by how the blast is triggered [30]. The former method categorizes these injuries as primary, secondary, tertiary, and quaternary. In general, primary injuries occur when the blast wave travels through the body [20], transferring energy to interfaces between tissues of different acoustic impedance (related to density), causing soft tissue destruction [8,20,23]. Stress waves propagating through a limb travel at approximately the speed of sound, causing microfractures [31]. The secondary injuries are caused by objects energized by the explosion, turning them into projectiles or shrapnel. The fragments that can be projected act under a spectrum of ballistic forces are varied, have differing geometry and are classified as primary or secondary fragments [23]. Primary fragments are flying debris from the explosive device, and the secondary ones are objects picked up by the blast wave and propelled in its wake [23]. Secondary injuries would include comminuted ballistic trauma (in which the bone is impacted at high speed, causing it to break into many fragments) or projectile (penetrating or perforating) trauma [20,32]. Ramasamy [20] incorporates a classification that he calls “mixed orthopedic injury”. This brings together primary and secondary effects, since the effects of the blast wave and the products of the detonation occur almost instantaneously. Tertiary injuries, also associated with the blast wave, occur when the victim is thrown against the ground or an object [29]. That is, tertiary trauma refers to blunt force trauma [20]. On long bones, lesions of this type are characterized by their butterfly shape [33]. Quaternary injuries represent a diverse category of injuries (including burns) and are not directly caused by the blast wave [20,29]. Rankin et al. [29] add a fifth categorization, which includes non-blast effects that lead to a hyperinflammatory state, including those caused by the use of biological, chemical, or nuclear products. Coupland and Korver, 1991, and Smith et al., 2017 [30,34], classify them into patterns, according to the way the explosion is triggered. Pattern 1 occurs when stepping on the mine causing serious lower extremity injuries, including traumatic amputations. Pattern 2 is when the device explodes near the victim, and pattern 3 when the device explodes during handling. Our results show that the effects of the explosion on the part of the carcass are different according to the type of organ, although, in most cases, they are equally devastating. In the first place, the detachment and/or disintegration and projection of the viscera present in the abdominal cavity before the explosion is highlighted. Structures with gaseous content, such as the intestines, disintegrated and it was not possible to find any pieces of them on the experimental field. The literature indicates that the gastrointestinal tract is among the organs that are most susceptible to primary explosion injuries, because it is full of gas [35]. Only the bladder, with its ligaments, and the right kidney survived the explosion. Both remained in the abdominal cavity and without apparent changes to their macroscopic structure. The kidney had only been detached from its original position. This suggests that its retroperitoneal location would have played a protective role on both organs, preventing their detachment and projection outside the cavity, which remained intact. The spleen, a peritoneal structure positioned in the upper part of the cavity, protected by and fixed to a still incipient fragments of rumen, was also expelled from the cavity. Unlike the other expelled organs, part of it was recovered during the search. The left kidney (not retroperitoneal and located more centrally within the abdominal cavity), was also expelled or disintegrated. No trace of it could be found during the search. In other words, not only the position of the organs is a determining factor for the degree and/or type of effects caused by the explosion, but also the characteristics and density of each of them. Previous studies [35] with humans already considered this notion, indicating that the differences in the densities of the anatomical structures of the body make them susceptible to detachment, due to acceleration/deceleration forces. This would cause the tearing of the pedicles of the organs and the mesentery when there is a difference in inertia between the organ structure [35]. On the other hand, the liquid/gas content present in the organs [35] and their elastic recovery capacity against high pressures [36] would make them more or less susceptible to primary injuries from the explosion. At the orthopedic level, there was bilateral damage to both hipbones, open fractures associated with a large loss of tissue, closed fractures, and complete avulsion of the right foot (metatarsals and phalanges) in direct contact with the AP mine during the explosion. Open transverse fractures were observed in the right femur and tibia, and there was major destruction of the joint cavities on the same side. The left limb, on the other hand, presented minimal external damage to the joints and skin. However, internally, it presented separations/dislocations of the distal epiphyses (with a high degree of comminution) in most of the long bones, and the absence of growth plates. Previous studies could help to explain our results. On the one hand, it has been shown that the severe axial force (blast wave) caused by an underground explosion, transmitted directly to a limb, can cause the same bristling (shattering) effects on bones and comminuted fractures [20] as those found in our study. This vertical force would also lead to mutilation of the foot and lower leg [16,37] immediately or by surgical removal [38], although the mechanism of injury by which the blast leads to amputation trauma is not yet clearly understood [29]. Hull and Cooper, 1996 [31], showed that the blast wave alone is capable of generating fractures in a goat’s limb, when it is close to the point of explosion and not only when it is on top of the device at the time of the explosion. On the other hand, Trimble and Clasper [8] point out that the forces produced by the products of the detonation and the hot air would have local effects on the foot, while their effect on the opposite limb would be less (without amputation). The latter would cause tissue detachment along the facial muscle planes [19], which is what occurred in the amputated limb, perineal and genital areas in our study. Other authors indicate that detonation products and hot air would cause avulsion through the fracture site [8,31]. Our results are consistent with these last authors due to the presence of strongly adhering grass and the intense smell of burning in the amputated area, and the severe damage to the soft tissue, bone and joints at points close to the explosive source. In other words, our findings would be due to a combination of the blast wave, the projection of detonation products and hot air. However, although all these effects, previously reported, coincide with our results, several differences are seen when comparing the type of damage/effect and the areas where it occurred. It has been reported that the intensity of comminuted fractures at the fracture point is related to the higher impact velocity [39,40], which is higher at the point of the explosion [41] and/or caused by the direct impact of a high energy fragment [42], for example, items propelled vertically upwards from the detonation point. It is also noted that more localized longitudinal loads in a mine lead to a higher incidence of distal fractures [8]. In our case, these effects were more evident in the opposite limb. Although the highest incidence of comminuted fractures occurred in the epiphyses of the distal regions of the long bones, they occurred in the limb that received the blast wave indirectly (laterally) and not directly on the bone axis, as occurred with the amputated right foot. However, given the absence of bony structures in the right foot, we cannot rule out the existence of comminuted fractures on the absent structures. Ramasamy et al. (2011) [20] indicated that amputation is seen as a short, oblique, or transverse fracture through the shaft of the long bone. Others [8,31,32,35] point out that there are specific bony points that are more prone to amputations, such as the proximal area of the tibia (at the level of the tibial tuberosity). In our study, it was not possible to observe fractures at the point of the traumatic amputation of the right foot. These and other differences relate to the type of effect, and the areas where they were established could be related to the age of the part of the animal carcass used in the study, as previously mentioned. Most reports and descriptions of injuries caused by antipersonnel mines refer to adults, in whom there is advanced bone development. When a living being is born, only part of its skeleton is ossified. The growth plate or physis is responsible for the longitudinal development of long bones [43]. This is a hyaline cartilage formed by type II collagen. A considerable amount of this is located between the diaphysis and epiphysis of the long bone, and its ossification, in bovines, occurs months after birth [43]. Long bones in newborn animals are much less resistant to impact than adult bones [44]. Already formed bones have type I collagen that allows them to withstand a high load [45]. Cartilage, on the other hand, formed by type II collagen, is particularly sensitive to mechanical loads due to its ease of deformation under continuous loads [46]. Both structures, therefore, have different densities [44]. On the one hand, the difference in densities between these structures would explain the effects found, according to Ramasamy et al., 2011 [20]. On the other hand, the lower structural resistance of part of the carcass we used vs. the healthy bones of adult individuals analyzed in previous articles could explain, at least in part, the differences in the type of fractures found and the bone regions where they occurred. Although we have already explained that our results show effects and not injuries, we consider an extrapolation valid in order to classify the magnitude of the damage. According to the authors’ classification for humans indicated above, the effects found in our study could be categorized mainly as primary and secondary lesions. Although the antipersonnel mine used in the study lacked materials inside that would act as shrapnel, small fragments of it could be projected at high speed and especially onto the body part in direct contact with the explosive device. These fragments could cause tissue damage before they disintegrate. Trimble and Clasper [8] point out that environmental remains and, to a lesser extent, the minimal metallic components of a mine produce penetrating wounds in the contact limb, the contralateral limb and the perineum. The detachment of tissues would open the way for the penetration of foreign objects [19], such as soil, grass and even a piece of fragmented bone that could act as a secondary projectile [16]. This effect was more evident along the right limb to the base of the tail, possibly as a consequence of the accumulation of energy at the point of detonation and the vertical projection of fragments originating from under the amputated foot at high speed. It has been indicated that the layer of soil and/or soil particles that covers a landmine is expelled at high speed [19], favoring its local effect and in areas close to the point of detonation [29,41]. According to this, the bones of the right foot could also participate in the generation of the effects found, prior to their disintegration. Some characteristics of the projected elements may also have an influence, as indicated above. Due to the low weight of soil particles and grass fibers, we believe that their presence could be a consequence of tissue shedding, rather than the cause of the damage, although it is impossible to totally rule out their contribution. The shape of the projected fragment(s) [23] could explain, at least in part, the damage caused to the left limb, without external injuries. It is possible that the flat surface of a fragment of the artifact struck the left limb, causing fractures, without penetrating injuries. In addition, it must be considered that the magnitude and dispersion of elements will depend on the type of soil, the presence of stones, and the depth of burial of the mine [8,19]. 

It is impossible, in our study, to differentiate primary effects from secondary or mixed ones. Given the existence of tissue damage with and without soil fragments or other embedded fragments, and the presence of a fragment of the explosive device in the field of experimentation, we could infer that both effects are present. However, the presence of fractures without penetrating injury in the limb opposite the point of explosion, the absence of embedded elements in areas of the part of the carcass far from the point of explosion, the absence of external damage to the abdominal cavity, and the expulsion of the viscera, incline us towards the hypothesis that most of these effects were caused mainly by the blast wave, followed by the action of fragments. Considering the type and composition of the soil is important during forensic investigation in cases of explosions. A correspondence must be established between the characteristics and severity of injuries and/or effects identified and the nature of the elements projected from the ground. The type of injury/effect and/or the presence of some vestige, projected onto or embedded in a body, could allow us to know or corroborate reports and testimonies about the characteristics of the scene of the events, when we were not present at the scene of the crime. In addition, we must not forget that those penetrating injuries caused by projectiles propelled by the explosion can cause infections as they bring bacteria into the interior of the body, which increases the risk in a living individual [23]. Neither can we rule out tertiary or quaternary lesions a priori. The absence of butterfly-shaped damage to long bones, reported by other authors [33], and the presence of bilateral damage, encourages us to rule out significant blunt force damage. The absence of the growth plates in the long bones or coxal cartilages, not yet ossified, could strengthen our hypothesis. The absence or disappearance of these cartilages was determined by not finding a clear correspondence when trying to connect the separated epiphyses with their corresponding diaphyses, or to articulate the coxal bones. Thermal injuries (quaternary effect) are common in any close-range explosion and a charge buried at ground level generates a high and sustained initial vertical wave velocity, allowing hot unburned products to mix with fresh oxygen and maintain the combustion process [19]. The concentration of energy at the point of detonation for this type of soil, for the reasons indicated above, could explain the lack of heat in areas more distant from the point of explosion. Moreover, the absence of incandescent fragments and/or visible burns on the casing allows us to infer that the explosive quantity was very small or that, due to the close proximity of the explosion, its effects were overshadowed by the other effects of the explosion. 

### 4.3. Work at the Crime Scene or Investigation Site

One of the first actions that a first responder or the investigation body (CSI team) must carry out in the field is to determine the primary (central) and secondary focus of the scene, in order to establish the investigation perimeter [22]. Next, the presence of evidence that can disappear quickly (temporary and transient effects) but that could constitute physical evidence must be recognized [22]. In our case, smoke, fertilizer odor, tar, noise, and vibration were temporary effects that needed to be recognized upon arrival at the scene, if possible, and documented immediately. The presence of the crater, evident more than 40 m away, allowed us to identify the central element of the scene. For specific cases of scenes where an explosion occurs, Thruman [9] suggests a multilevel perimeter system that defines three areas isolated from each other. The first level, or internal perimeter, should include a focus on the explosion and/or where the evidence is found. This first limit should be set at one and a half times the distance from the epicenter of the explosion to the last evidence found. The second level or external perimeter is established at a distance that prevents non-investigative personnel from observing the work on the scene or hearing what the investigators may discuss. In our case, the last evidence was found more than 40 m away, almost at the limit of the area surrounding the field of experimentation. That is, we should set the internal perimeter at 40 m and the external one at 60 m from the center of the crater. However, in our case, the topography of the location and the presence of light constructions (e.g., greenhouses) did not allow the installation of an artificial external perimeter at the indicated distance. The topographic factors determine how to establish the perimeter [9]. That is, the perimeter can be established artificially, through the use of cones, tapes, fences, structures already present in the place (e.g., building), or natural barriers [9,22]. These perimeters create an area between the focus of the detonation and the last fragment of evidence. There is a second zone between the internal and external perimeter, and a third zone outside the external perimeter. This multi-level perimeter is suggested for many crime scenes, from a functional point of view, allowing the establishment of specific work points for the different teams present at the scene [9,22]. The explosion of a home-made anti-personnel mine and the impact caused by the detonation should constitute a warning of the possible presence of other devices of this nature in the area. The risk of these artifacts is high due to the impossibility of detecting them by traditional methods, because of the absence or low quantity of metallic elements used in their construction [10]. In summary, it is necessary to evaluate all security aspects and use the best criteria to determine the perimeter of protection. On this basis, we can point out that the concept of perimeters particularly refers to an approximate distance away from the main focus of the scene that allows safe work. In our case, the hills constituted our outer perimeter, after verifying that there was no visible evidence beyond them. A natural perimeter should not be established by the mere fact of being present at the scene since there may be evidence beyond it or it may provide circumstantial evidence of the case, such as allowing or preventing the escape of suspects [9,22]. During the search, the best search procedure was found to be the zonal pattern. Given that each area, of 4m^2^, was examined by a single investigator, this allowed them to find a greater number of evidence (especially particles of soil) and caused less destruction of others not recorded by the other methods. Some of the factors that influence the search at the crime scene are the brightness, the size of the object, the capabilities of the searcher and the characteristics of the terrain [9,22]. The latter seemed to be the most influential factor in our results. All those who collaborated in the search for evidence pointed out that the abundant vegetation made the process difficult. Given the abundant vegetation, many particles of soil scattered in the field were crushed when using the other search patterns and were only registered by the zonal method. Despite the advantageous result of the zonal search, it turned out to be much slower than the other procedures. In fact, the search team did not manage to cover the entire area in the time defined for the experiment. However, this search, more exhaustive and meticulous than the others, would allow, theoretically, the reduction of the risk of stepping on and detonating another device by the investigation personnel and, incidentally, contribute to the demining process. Having several investigators should theoretically increase the concentration span of each of them, and the exhaustiveness of the search should decrease the levels of anxiety in the rest of the team in the face of another accidental explosion. Searching in lines and stripes with multiple browsers was second in terms of efficiency. This allows us to assume that the number of search methods is vital for a more efficient result. However, second place in the “efficient search ranking” increases the probability that one of the investigators could detonate a new device, endangering the entire team. Individual searches were further affected by vegetation. The linear and individual grid search did not differ much from spiral search. However, it was more orderly. One of the disadvantages of the spiral search, already discussed by other authors, is the difficulty of maintaining the same distance between each turn and the speed rhythm [22], which was very evident in our study. With the exception of the zonal pattern search, the search was completed using the other methods in the entire defined study area and in the designated time. Completing the search in a timely manner depended on each investigator or group of investigators. The constant risk of the presence of other artifacts should play an important role in the time and manner of the search, along with the factors already mentioned above. In our case, it was known that there was no other artifact in the experimental field. The question is: in a real case, how much could this uncertainty influence the search efficiency? Some experts [9,22,47] point to the need to define a single point of entry and exit to the crime scene, which provides a means of control over who enters or leaves the scene. We can add the need at the beginning of the investigation to create a safe and direct entry and exit route to and from the detonation point to determine early aspects related to the explosion (e.g., type of explosive, inherent risks, etc.) that help in research planning. This route must avoid destroying any evidence or putting anyone using it at risk. Risks can be present upon arrival, or appear at any time, so safety must be constantly evaluated [22]. This same access route can become the initial point of search, as in this study. Thurman [9] places it in the interior perimeter as a means of control. However, authors such as Gardner and Krouskup [22] point out that in a multilevel perimeter, there can be as many perimeters as the investigator decides, as long as they are functional, and each of them can have a single entry and exit point. Once again, the criterion and experience of the researcher will be of great importance. Our study demonstrates that the effects on the part of the carcass and the logistical challenges encountered in investigating a scene where a homemade antipersonnel mine explosion has occurred are diverse, unpredictable, and enormously destructive. Despite being a small artifact, inferior in capacity and construction to those causing reported injuries in humans, it was capable of causing similar patterns. It is evident that the effects caused by the explosion are multifactorial and include the characteristics of the explosive device, the potential victim and the environment, and the degree/type of relationship that is generated between them. It is clear that any permutation between these variables is possible [14,19,24,35,39]. Our findings could contribute, at least in part, to demonstrating possible scenarios relating to the magnitude of damage that this type of explosive device could cause to humans and living non-human animals that share similar anatomical characteristics to the part of the carcass we used, in relation to its proximity to the explosive device during the explosion, and ruling out possible vital tissue reactions present in a living animal as an *ante-mortem* effect. The poor musculoskeletal development of our animal model, due to its age, may be analogically and homologically similar to other species of mammals (included human beings) who are still exposed to these active explosive devices. One of them is the degree of ossification of growth plates, which decreases towards puberty [48]. It also demonstrates the risk that researchers face. The homemade manufacture of these antipersonnel mines makes them undetectable [4,10,11]. Even with minimal amounts of explosive, they are transformed into an unpredictable public enemy, causing incalculable damage (some irreversible) to the fauna and the population in general, directly or indirectly, in Colombia and numerous places in the world [3,7,49,50,51,52,53,54]. All this knowledge can contribute to improving the emergency treatments that must be used for victims (human and non-human animals) of this scourge. The number, the type of animals (alive and dying), and the degree of injury that they present at the scene can be highly variable and must be evaluated in situ, efficiently, to identify seriously injured patients who require immediate attention (Triage) and provide them with the necessary care to save their lives and transport them to veterinary centers with the capacity to care for them. For this, it is imperative to have personnel trained to prioritize emergencies at the scene and provide adequate attention in care centers. All this must be combined with coordinated action with other professionals and care centers. Effective triage and patient distribution is vital, requiring system planners to take into consideration both the experiential basis of past events and the limited evidence available for triage methodology in these types of cases, and be able to adapt their local plans accordingly. We must not forget the possible post-traumatic stress disorder (PTSD) that traumatic events of this type can cause on animals that could be studied. Improving the medical care of patients in these events requires a broad understanding of the epidemiology of injuries caused by this type of explosive device, which aim more specifically at maiming than killing, as we pointed out at the beginning. From the point of view of the field investigation, recognizing evidence caused by a possible explosion early, during the ocular technical observation, requires taking intense security measures. This should include the safekeeping of research equipment, equipment, etc. The entrance by veterinarians and the investigation team must be carried out after professionals specialized in explosives have determined the safety of the site. The problem, as we pointed out above, is that these artifacts are not detectable by conventional means, and the only way to safeguard the integrity of personnel and evidence is to make safety a continuous task for everyone. This requires good planning, together with specialized professionals who have knowledge in this type of event. Our results show that patterns commonly used in this type of events do not seem to be the most appropriate since the researcher is repeatedly exposed to other possible artifacts planted in the sector, or the environment makes it difficult, etc. The fact that the most appropriate search pattern is the one that uses the greatest number of personnel and takes the most time, may be the result of the lack of knowledge that forensic veterinarians have on this subject, and being “fearful” leads to “excessive care”. The positive aspect of this “excess of care” is that it alerts future researchers to the risks they may face, and reminds knowledgeable researchers that overconfidence is a risk. We must not ignore the fact that sometimes what seems to be the “most appropriate” could hinder the fluidity of the investigation in the field, unnecessarily increase the time of exposure to new artifacts, and increase the stress of the research team. Experience then allows us to point out that sometimes what is planned, before arriving at the scene, is not possible to carry out fully, and one must act painstakingly on the basis of the experience that a forensic scientist (e.g., a forensic veterinarian) must acquire, together with other more highly trained professionals at the crime scene. Knowing the epidemiology of these artifacts, as we pointed out above, can also help prevent or reduce vulnerability to them through the use of more effective personal protective equipment, designed on the basis of knowledge of this type of artifact, that are built with easily obtained components. 

The mixture of ammonium nitrate and aluminum used as an explosive is well-known [55] and the use of ammonium as a fertilizer makes it attractive as an explosive due to its low cost and ease of procurement [4], and this also hinders the control of its sale and distribution. The number of explosive substances used by armed groups in these homemade devices is a fundamental factor in the post-explosion effects. In fact, the capacity of large amounts of ammonium nitrate as an explosive is already known, at the population level, after the explosion in an urban area in Beirut, which was broadcast to the whole world [14]. 

## 5. Conclusions

The explosion of a homemade anti-personnel mine, without shrapnel inside its container, showed behavior similar to that of other conventional and/or improvised explosive devices. The blast wave, the projection of possible fragments, and the heat generated by 100 g of explosive substance R1, were variable enough to cause devastating effects, similar to those reported in previous studies with devices of greater capacity. It is difficult to determine which mechanism was predominant, although the dimension of the effects proved to be dependent on the distance from the source of the explosion, which explains the different patterns of damage between the limbs and other areas of the part of the carcass. 

New empirical studies and analyses based on real cases are necessary, involving animals affected by explosions of this type of device, to help improve emergency clinical care in human and veterinary clinics.

This new knowledge will make it possible to improve investigation procedures at scenes of this nature, promote the formal development of animal forensic science, and open up new possibilities for multidisciplinary work.

New empirical studies must be added, and new alternatives for humanitarian demining promoted. More broadly, there must be investment in educational projects that would allow the population to understand the true scope of the problem in order to prevent new and painful disasters. Future studies must also include new steps, such as documentation (e.g., photography, video recording, notes, sketches, etc.), collection, conservation, preservation and transfer of evidence to a laboratory (under a chain of custody), and clearing of the scene before its release, among others. 

The dimension of the damage to biodiversity and the constant risk to investigators dealing with the scenes of explosions of home-made antipersonnel mines makes it essential to create public policies that will allow resources to be invested in adequate logistics, tools, and human capital specialized in criminal investigation in the countryside, e.g., veterinarians. 

## Figures and Tables

**Figure 1 animals-12-01938-f001:**
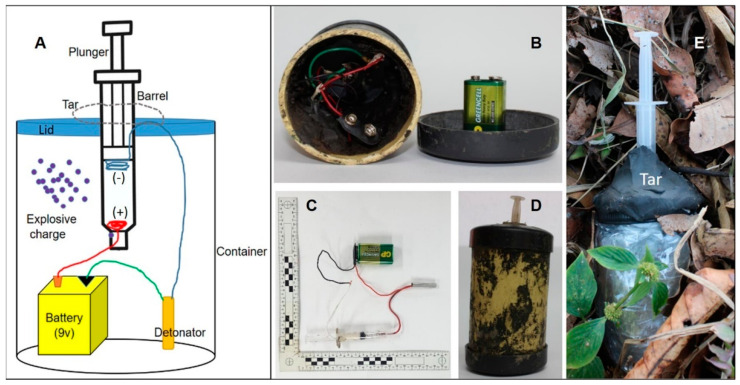
(**A**) Design of the homemade explosive device used in the study and layout of its components; (**B**) Internal arrangement of the device without the explosive charge and power source not connected; (**C**) Connection between the electrical circuit, power source and fuse; (**D**) The explosive device before activation and sealing; (**E**) Complete assembly of the waterproofed homemade AP landmine.

**Figure 2 animals-12-01938-f002:**
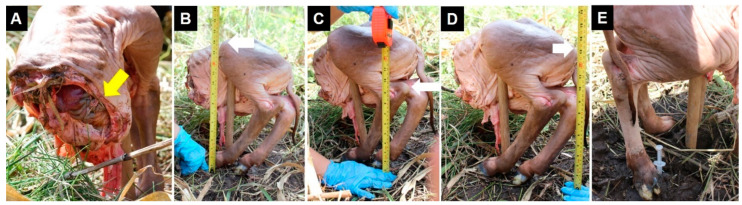
(**A**) The yellow arrow shows some visceral, organs and tissues still present in the abdominal cavity; (**B**) The distance between the explosive and the coxal tuberosity (high: 36–37 cm) (white arrow); (**C**) The distance between the explosive and the calcaneal tuberosity (high: 23–25 cm) (white arrow); (**D**) The distance between the explosive and the ischial tuberosity (high: 32–33 cm) (white arrow); (**E**) The homemade AP landmine positioned by the inner side of the animal’s right foot.

**Figure 3 animals-12-01938-f003:**
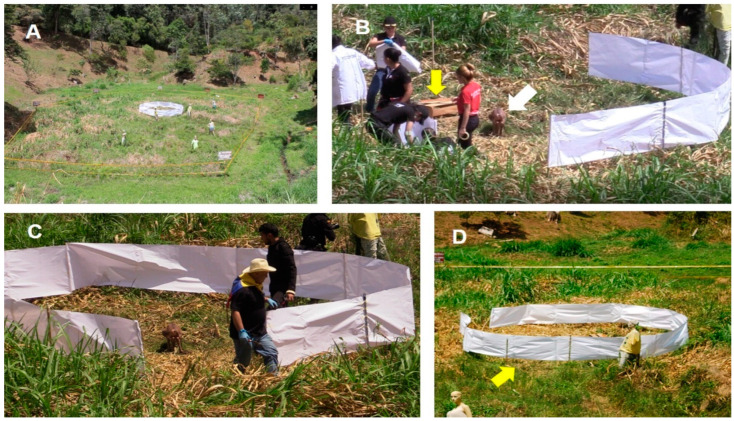
(**A**) Panoramic view of the experimental field bounded by a yellow tape; (**B**,**C**) south view; (**D**) east view. (**B**) Wooden box (yellow arrow) to the left side of the animal carcass (white arrow); (**C**) Position of part of the carcass in the center of the perimeter. Only a small portion of the fence remained open for a better view of the epicenter of the explosion; (**D**) Panoramic view of the paper fence before the explosion. Free space below the fence (yellow arrow).

**Figure 4 animals-12-01938-f004:**
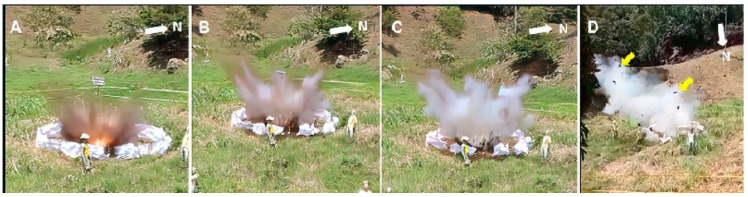
Video snapshots during the explosion of the homemade AP landmine. (**A**–**C**) Recording from the east sector of the experimental field. (**A**) In the center of the explosion it is possible to observe a “fireball” (exothermic reaction) while a column of greyish smoke, is projected up and sideways over the fence. The latter begins to disintegrate; (**B**) A few hundredths of seconds after the explosion, the light disappears while the grayish smoke column continues to expand. Major effects on fence are evident; (**C**) The smoke is whiter in color than at the beginning of the explosion; (**D**) Recording from the north sector of the experimental field. The smoke moved eastward along with the wind direction. Some soil particles were projected towards this sector (a few of them are marked with a yellow arrow). Figure 5F shows the projection of solid elements into the western sector of the field. The elements projected towards the southern and eastern sector of the field are not shown; (N) North view.

**Figure 5 animals-12-01938-f005:**
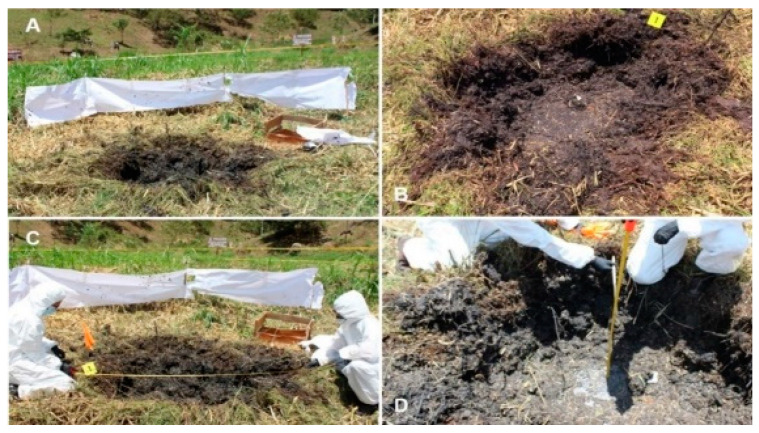
Effects of the shock wave on elements present in scene. (**A**) General view of the scene; (**B**) Crater; (**C**,**D**) Crater diameter and depth measurements, respectively; (**E**) Photograph showing the effects on the wooden box and its spatial relationship to the crater; (**F**) Part of the carcass ascending after the explosion (dotted yellow circle). It is possible to visualize dark solid particles moving in the direction of the camera (east position) in the smoke; (**G**,**H**) Degree of destruction of the fence registered from southern and eastern positions, respectively.

**Figure 6 animals-12-01938-f006:**
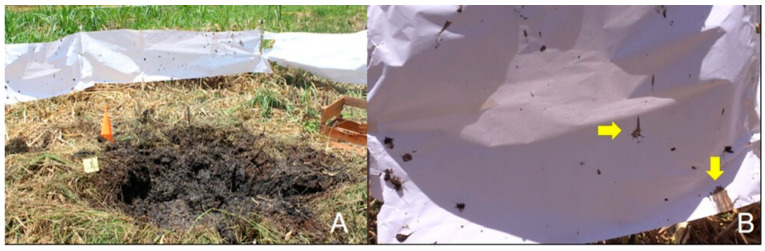
(**A**) Overview of the scattering patterns of the blast-driven particles of soil on the fence; (**B**) Close-up view shows different patterns of humid soil particles on the fence (yellow arrows).

**Figure 7 animals-12-01938-f007:**
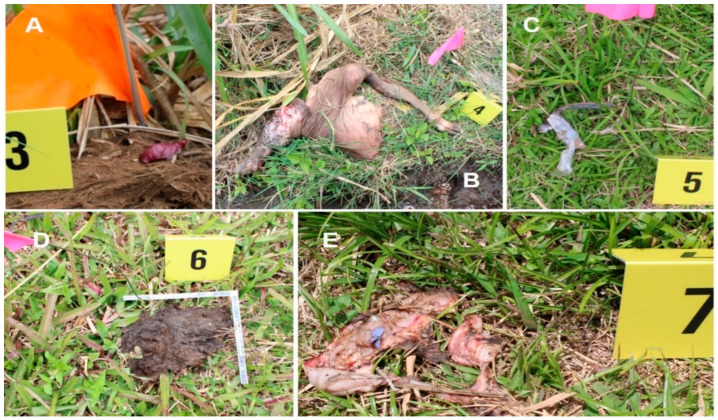
Evidences. (**A**) Fragment of spleen; (**B**) A large portion of the casing; (**C**) Container fragment wrapped in waterproofing foil; (**D**) Particles of soil (other particles of soil projected from the explosion point are not shown); (**E**) Fragments of soft tissue.

**Figure 8 animals-12-01938-f008:**
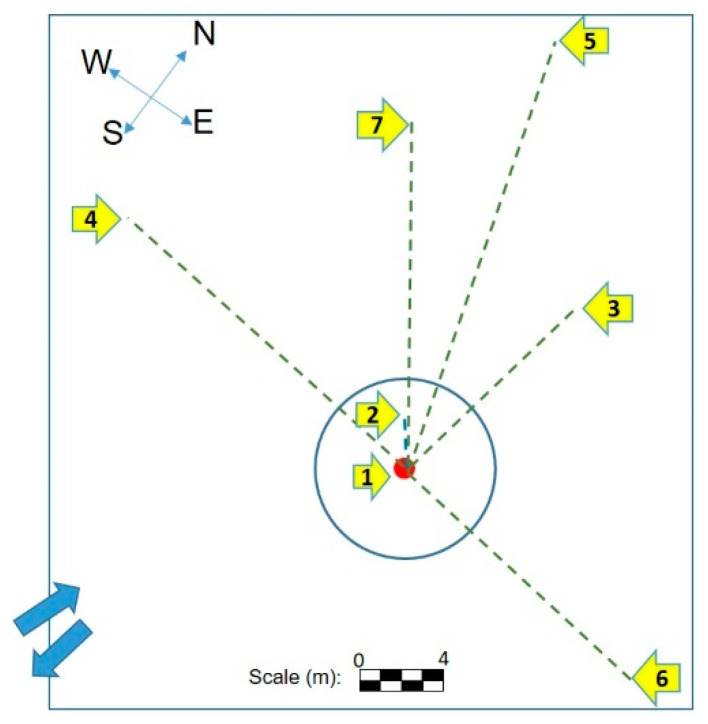
General schematic presentation of the distribution of evidence found after the explosion following an eccentric spiral research pattern. The number designated to each evidence is displayed in yellow arrows, and the green and blue dotted lines show the distances measured. (1) Crater epicenter (red circle); (2) Wooden box; (3) Fragment of spleen; (4) Largest portion of the part of the carcass; (5) Fragment of the AP landmine; (6) Particle of soil; (7) Soft tissue. The blue circumference represents the fence located at a radius of 4 m around the homemade AP landmine. Only one particle of soil was numbered and photographed for didactic purposes. The blue arrows correspond to the entry and exit points to the scene.

**Figure 9 animals-12-01938-f009:**
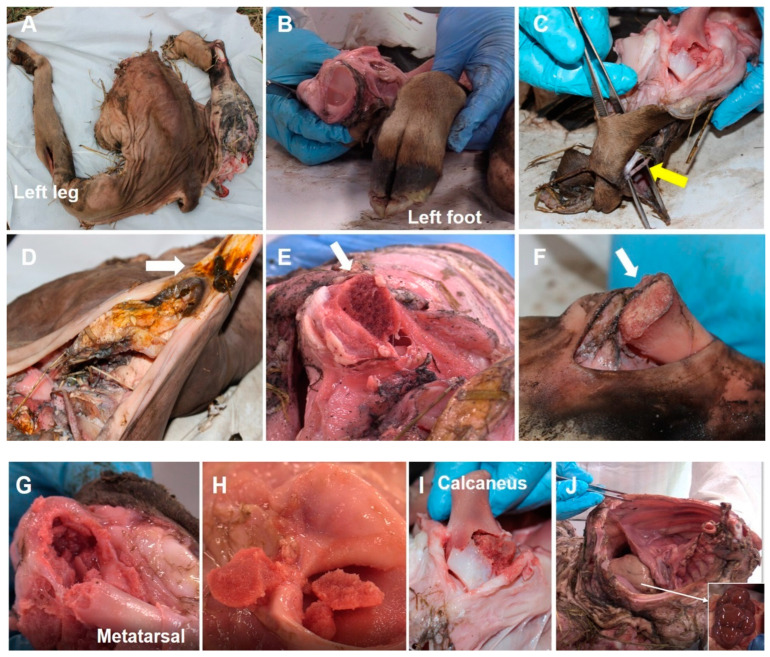
(**A**,**B**) Showing severe mine blast injuries (effects) in the experimentation field and laboratory, respectively. Comparison of complete amputation of the right foot and survival of the left foot; (**C**) The effects of exposure passed through the fascial planes giving rise to the classic umbrella effect of the remaining soft tissues. It is even possible to see pieces of tendons and nerves still present; (**D**) Destruction of the facial planes in the perineal and genital zones; (**E**) Transversal fracture of the proximal metaphysis of the right femur (white arrow); (**F**) Transversal fracture of the distal metaphysis of the right tibia (white arrow). In the small lower box, luxation of the distal epiphyses of the left tibia; (**G**) Luxation of the distal epiphyses of the left metatarsal bone; (**H**) Comminuted fracture on the dislocated distal epiphysis of the left metatarsus; (**I**) Fracture of the calcaneus; (**J**) Abdominal cavity without viscera after the explosion. Only a depleted urine bladder with ligaments and the right kidney (out of its normal position) are still present in the abdominal cavity. The small lower box shows, the external integrity of the right kidney.

**Table 1 animals-12-01938-t001:** Summary of the distance of each piece of evidence to the crater center.

Evidences	Distance from Crater (m)
Wooden box	2.15
Fragment of spleen	9
Largest portion of the part of the carcass	14.4
Fragment of the AP landmine	19.4
Soft tissue	15.6
Particle of soil expelled from center of the crater	13

## Data Availability

Not applicable.

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
