# Peer review of "Preliminary Considerations for Crime Scene Analysis in Cases of Animals Affected by Homemade Ammonium Nitrate and Aluminum Powder Anti-Personnel Landmines in Colombia: Characteristics and Effects"

_animals, 2022, doi:10.3390/ani12151938_

Round 1

Reviewer 1 Report

General Structure and comments

The paper is well structured, following the objectives. As major general comment could be pointed a large discussion that, perhaps, could be more synthetic. The following comments will address considerations/suggestions that are more specific.

Specific comments

Ln46 – in fact the work have 3 objectives (describe the characteristics of a controlled explosion of a homemade antipersonnel landmine; compare the effectiveness of some evidence search patterns used in forensic investigation; determine the effects on a piece of an animal carcass) and not only one;

Ln116 – Water permeability is, in this case, a soil characteristic, and could be low, high,…. The water flow through the soil could be slow or fast. Consider clarifying;

Ln131 (Ln770) – consider give more information about the reference (thesis, dissertation, personal communication?). It was not possible to track it, even in the Medellin University site;

Ln137 – percentages are not given. Note that for the reasons pointed, it is not needed. Consider phrase correction;

Ln147 and following – The reason for not required an ethical review and approval is not related with the use of a part of a carcass nor the anatomical components present, but the fact that the animal died of natural causes and the body (part of) was donated. So no live animal was use or sacrificed for the experiment. Consider rewrite to clarify;

Ln154 – consider using anatomical planes (cranial instead of front);

Ln158-161 – consider to move text to the description of the device;

Ln174-179 – consider to give more detailed information about the “analysed variables”. Not only “what” was consider as variable, but also “how” the variables were measure/analysed. This also will make the text of the discussion more fluid, with no need to be back to the methods. You also may want to consider that some variables were not really analysed but just empirical observed/recorded Consider also to harmonize the terms “animal”, “carcass” and “piece/part of carcass”. In reality, the experiment do not use a carcass, but just a part of it, and that could influence the results/discussion (see below);

Ln219 (figure 4) – as the search patterns considered, are usual in crime scene investigation, means that they are not developed in this research, the figure 4 could be downsized or even be acknowledge only by introducing a bibliographic reference;

Ln234 – “a higher intensity of fertilizer with the smoke”. Need clarification. Smell, odor?...

Ln264 – “… more intact…”. Intact is an absolute value. Consider rephrasing, e.g. with less damage;

Ln321 (figure 8) – (F) although this information could be very interesting, it is not addressing/developed in the discussion, so you may want to consider if it is important/relevant for the present work/photo gallery;

Ln330 (figure 9) – sketches are an important tool in the crime scene investigations and, during the field collection of evidences they usually are not drawn to scale. But, for the present paper consider to present it on scale, as it was part of an expert report.

Ln388 and 389 – there is no “small lower box” in plates (E) and (F);

Ln490-491 – although odor could be of capital importance to rise suspicious about several compounds, it seems radical to state that it would be useful to “determine qualitatively” which components are present;

Ln494 and following – in the item 4.2 of the discussion, it would be useful to discuss, right at the beginning, two major features of the animal model: 1) is a “half carcass” and 2) is a “stillborn”. Note that the fact of an “open ended” (cranial aspect of the piece of carcass) could be relevant in all the discussion about effects, at least on viscera. On the other hand, the fact of being a stillborn could be relevant in the discussion (line 625-626) about butterfly fractures, since the bones of animals of this age, as the authors acknowledge, could present more elasticity, and behave differently to tension/compression forces. It was noted that the authors addressed the discussion of the age (ln575-576), nevertheless should be considered to introduce this issue earlier in the discussion;

Ln558 – the results of a research are (or not) consistent with previous research(s) and not the other way around;

Ln604 – consider change “we are inclined to think” with a more “scientific sound like” phrase;

Ln642-643 – in this case, or in others, seems difficult to “record upon arrival at the scene” the smoke, noise or vibration, because, as the authors state, these are temporary effects. Consider rewriting the sentence for clarification;

Ln714 – extrapolation to children, based on the weight/height/size of the animal model used in this experiment, should be done carefully, because there are other variables that are not taken in to consideration and may have relevant impact in the outcome;

Ln727-738 – the text, although important, is not a discussion of results. You may consider to move it elsewhere, namely to conclusions.

Reviewer 2 Report

This study is well organized and documented.  Editorial comments are included in attached pdf document.  

The conclusion statement should be reworded.  As it currently reads:

"New empirical studies and analyses based on real cases are necessary, 746 involving animals affected by explosions of this type of device, to help improve emergency clinical care in human and veterinary clinics. The new knowledge will make it possible to improve investigation procedures in scenes of this nature, promote the formal development of animal forensic science, and open up new possibilities for multidisciplinary work."

These are two very broad aspects, and each should be expanded upon.  The idea of "improve emergency clinical care in human and veterinary clinics" involves medical sciences.  The authors should state exactly how this type of research will improve medical (and veterinary) care.

Additionally, "The new knowledge will make it possible to improve investigation procedures in scenes of this nature".  The link to the forensic sciences is clear in the manuscript.  The authors should better summaries how, exactly, this study will improve crime scene investigation.  A better summation of their work will make a stronger conclusion for this research. 
